# Neighborhood cohesion and violence in Port-au-Prince, Haiti, and their relationship to stress, depression, and hypertension: Findings from the Haiti cardiovascular disease cohort study

Lily D. Yan[1,2]*, Margaret L. McNairy[1,2], Jessy G. Dévieux[3], Jean Lookens Pierre[4], Eliezer Dade[4], Rodney Sufra[4], Linda M. Gerber[5], Nicholas Roberts[1,2], Stephano St Preux[4], Rodolphe Malebranche[6,7], Miranda Metz[1,2], Olga Tymejczyk[8], Denis Nash[8], Marie Deschamps[4], Monica M. Safford[1], Jean W. Pape[2,4], Vanessa Rouzier[2,4]

1 Division of General Internal Medicine, Department of Medicine, Weill Cornell Medicine, New York, New York, United States of America, 2 Center for Global Health, Department of Medicine, Weill Cornell Medicine, New York, New York, United States of America, 3 Department of Health Promotion and Disease Prevention, Robert Stempel College of Public Health and Social Work, Florida International University, Miami, Florida, United States of America, 4 Haitian Group for the Study of Kaposi's Sarcoma and Opportunistic Infections (GHESKIO), Port-au-Prince, Haiti, 5 Department of Population Health Sciences, Weill Cornell Medicine, New York, New York, United States of America, 6 Collège Haïtien de Cardiologie, Port-au-Prince, Haiti, 7 Medicine and Pharmacology, Université d'État d'Haïti, Port-au-Prince, Haiti, 8 City University of New York Institute for Implementation Science in Population Health, New York, New York, United States of America

* liy9032@med.cornell.edu

**Data Availability Statement:** Data contain potentially identifying and sensitive patient information. Deidentified data used for this analysis

## Abstract

Neighborhood factors have been associated with health outcomes, but this relationship is underexplored in low-income countries like Haiti. We describe perceived neighborhood cohesion and perceived violence using the Neighborhood Collective Efficacy and the City Stress Inventory scores. We hypothesized lower cohesion and higher violence were associated with higher stress, depression, and hypertension. We collected data from a population-based cohort of adults in Port-au-Prince, Haiti between March 2019 to August 2021, including stress (Perceived Stress Scale), depression (PHQ-9), and blood pressure (BP). Hypertension was defined as systolic BP $\geq$ 140 mmHg, diastolic BP $\geq$ 90 mmHg, or on antihypertensive medications. Covariates that were adjusted for included age, sex, body mass index, smoking, alcohol, physical activity, diet, income, and education, multivariable linear and Poisson regressions assessed the relationship between exposures and outcomes. Among 2,961 adults, 58.0% were female and median age was 40 years (IQR:28–55). Participants reported high cohesion (median 15/25, IQR:14–17) and moderate violence (9/20, IQR:7–11). Stress was moderate (8/16) and 12.6% had at least moderate depression (PHQ-9 $\geq$ 11). Median systolic BP was 118 mmHg, median diastolic BP 72 mmHg, and 29.2% had hypertension. In regressions, higher violence was associated with higher prevalence ratios of moderate-to-severe depression (Tertile3 vs Tertile1: PR 1.12, 95%CI:1.09 to 1.16) and stress (+0.3 score, 95%CI:0.01 to 0.6) but not hypertension. Cohesion was associated with lower stress (Tertile3 vs Tertile1: -0.4 score, 95%CI: -0.7 to -0.2) but not

are available upon request after signing a data access and use agreement, provision of approval by the GHESKIO ethics board, and demonstration that the external investigative team is qualified and has documented evidence of human research protection training. Requests may be addressed to authors or irb@med.cornell.edu.

**Funding:** Funding for this study comes from the National Heart, Lung, and Blood Institute, grant numbers R01HL143788 (MLM, JGD, JLP, MHL, OT, DN, MD, JWP, VR), R01HL143788-S1 (MLM, VR) and D43TW011972 (JLP, ED, RS). The funders had no role in the study design or execution of this protocol.

**Competing interests:** The authors have declared that no competing interests exist.

depression or hypertension. In summary, urban Haitians reported high perceived cohesion and moderate violence, with higher violence associated with higher stress and depression.

## Introduction

Neighborhood context can directly impact health through the physical environment, such as sanitation or access to healthy foods, and the social environment, including social support and interactions [1, 2]. Neighborhood-level social cohesion and violence are two important social environmental factors [2]. These environmental and social determinants of disease, as described in the social ecological model of health, can substantially increase a person's risk of developing poor mental and physical health outcomes, including stress, depression, and elevated blood pressure [3–6].

Perceived neighborhood collective cohesion, or the level of trust and attachment in the neighborhood, can lead to informal mechanisms by which residents provide social support or promote public order [1, 7]. Higher neighborhood cohesion has been associated with higher aerobic physical activity and lower smoking [8, 9], better self-rated physical health [7], and even lower mortality risk on a community level [10]. In contrast, perceived neighborhood violence may decrease community trust and weaken social bonds [11]. Exposure to neighborhood violence has been associated with increased alcohol and drug use in low- and middle-income countries (LMICs) [12], and increased hypertension and rates of myocardial infarction or stroke in high-income countries [13].

Haiti is the poorest country in the western hemisphere, with a long history of political strife, violence, and instability, coupled with recurrent natural disasters in Port-au-Prince which have weakened public infrastructure [14, 15]. Haiti also has a history of strong community bonds, social networks, and resilience [16]. Given the severity of Haiti's current political, economic and social environment, neighborhood effects such as perceived cohesion and violence are hypothesized to have an impact on health outcomes. Our prior work has suggested complex social and environmental factors that may impact healthcare access and utilization [17]. There remains a knowledge gap in describing these neighborhood social environmental factors and their association with mental and physical health outcomes in Haiti and similar LMICs [2].

In this analysis, we investigated the relationship between perceived neighborhood cohesion or perceived violence, and the mental health outcomes of stress and depression, and the physical health outcome of hypertension, in a population-based cohort of adults in Port-au-Prince, Haiti. The conceptual model for this analysis hypothesizes that socioeconomic determinants (low cohesion or high violence) may increase stress levels, depressive symptoms, and hypertension, which in turn contribute to high rates of cardiovascular disease burden and mortality.

## Methods

### Study design and population

We collected data from a cross-sectional enrollment survey within the Haiti CVD Cohort Study (clinicaltrials.gov NCT03892265), which uses multistage random sampling to follow a population-based cohort of Port-au-Prince residents, as previously described [18]. The sampling frame was created from census blocks as enumerated by the Institut Haitien de Statistique et d'Informatique, with exclusion of blocks experiencing political violence. Using Geographic Information Software, waypoints were randomly assigned across census blocks,

with number of waypoints per block proportional to its estimated population. Study staff then used standardized procedures to select the closest residential building to each waypoint within a 50-m radius to approach for screening. This larger study has enrolled 3,005 participants without cardiovascular disease and follows them for 2 to 3.5 years to evaluate the prevalence and incidence of CVD risk factors and diseases, such as hypertension, diabetes, obesity, dyslipidemia, kidney disease, poor diet, smoking, physical inactivity, and inflammation. This study is led by the Groupe Haitien d'Etude du Sarcome de Kaposi et des Infections Opportunistes clinics (GHESKIO), a medical non-profit organization that has operated continuously over four decades in Haiti to provide clinical care and conduct research on HIV and chronic diseases.

Fig 1 shows the study area sampled in the study, which includes the neighborhoods of Cité Soleil, Jalousie, Belair, Carrefour, and Delmas that range from slum areas with extreme poverty and temporary housing structures to low-middle class areas with individual houses. In slum communities in Haiti, a neighborhood is often composed of multiple smaller units called *lakou*, a common courtyard shared between households, where most of the cooking, cleaning, communal raising of kids, and toileting occurs between multiple families. These *lakou* promote strong social bonds and support systems for their residents [19].

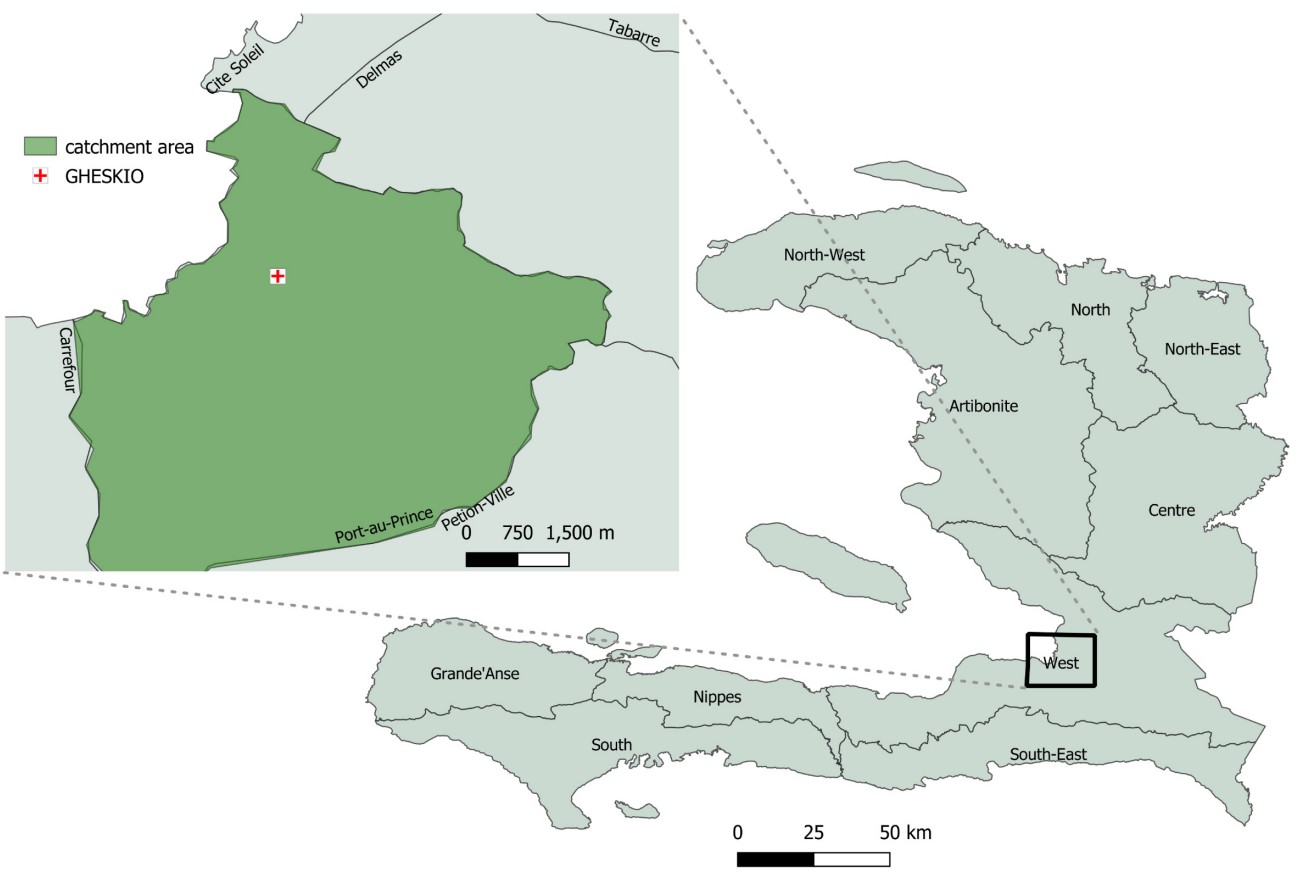

**Fig 1. Map of census blocks sampled in Haiti CVD cohort.** Map of sampled catchment area in Port au Prince. Base layer of map from Humanitarian Data Exchange, https://data.humdata.org/dataset/cod-ab-hti, CC BY IGO.

Inclusion criteria were age $\geq 18$ years, and primary residence in Port-au-Prince. This analysis included participants enrolled between March 2019 to August 2021. Participants missing data on neighborhood cohesion, neighborhood violence, perceived stress, depressive symptoms, systolic blood pressure (SBP) or diastolic blood pressure (DBP) (n = 44, 1.5%) were excluded (Fig A in S1 Text).

## Measurements

Sociodemographic data (age, sex, education, income) were collected using a standardized questionnaire at enrollment. We classified the level of education in two groups: primary school or lower versus secondary school or higher. Daily income was measured in Haitian Gourdes and converted into two categories of $\leq 1$ USD versus >1 USD. Health behaviors (diet, smoking status, alcohol, physical activity) were collected using standardized WHO STEPs instruments [20].

Smoking status was ascertained from questions asking if the participant ever smoked tobacco, and if they currently smoke tobacco. Alcohol intake was grouped into $\leq 1$ drink daily or >1 drink daily. Physical activity was determined from questions asking whether the participant did vigorous work or recreational activity for > 75 minutes a week, or moderate work or recreational activity for > 150 minutes a week. Participants were categorized as "moderate-high" if they reported yes to moderate or vigorous activity. We categorized diet by the median daily servings of fruit or vegetable intake, and categorizing it by < 5 servings a day versus $\geq 5$ servings a day (the recommended limit by the WHO) [20].

Clinical data including height, weight and BP were measured during a physical exam with a study nurse or physician at the time of study enrollment. Body mass index (BMI) was categorized using standard thresholds into underweight or normal (BMI $\leq 24.9$ kg/m$^2$), overweight (25.0–29.9 kg/m$^2$) or obese ($\geq 30.0$ kg/m$^2$).

## Neighborhood exposures

The main exposures of interest were perceived neighborhood cohesion and perceived neighborhood violence, measured using adaptations of validated questionnaires from the US, translated into Haitian Creole by GHESKIO staff and rechecked by a research staff fluent in English and Haitian Creole. Exact questions and scores used are detailed in Table A in S1 Text.

Neighborhood cohesion was measured using the Neighborhood Collective Efficacy Scale [1, 7]. The original instrument consists of 5 questions (e.g., "people in my neighborhood are willing to help their neighbors"), with answer choices based on a 5-point Likert scale (1 = Strongly disagree to 5 = Strongly agree), and questions 4 and 5 reverse coded (1 = Strongly agree to 5 = Strongly disagree). There is high between-neighborhood reliability and internal validity reported in prior literature [1]. Our adaptation retained all 5 original questions. The final neighborhood cohesion score was calculated by tabulating the individual question scores, for a total score ranging from 5 to 25, and Cronbach's alpha of 0.88.

Neighborhood violence was measured using the City Stress Inventory, Exposure to Violence subscale [1, 11], which consists of 7 questions (eg "A family member was attacked or beaten") and answer choices based on a 4 point Likert scale (1 = Never to 4 = Often). The original scale was internally consistent, stable, and correlated with US census indices of social disadvantage [11]. Our adaptation included 5 questions, modified with local study physician input for the Haitian context. The two questions "a family member was stabbed or shot" and "a friend was stabbed or shot" were combined for parsimony. The question "A family member was stopped and questioned by the police" was eliminated based on local Haitian input given reduced relevance in Port-au-Prince. A total score ranging from 5 to 20 was calculated by

adding up individual question scores, with a Cronbach's alpha of 0.44. Neighborhood cohesion and violence scores were kept on the individual level (not aggregated), and thus represent the perceived neighborhood cohesion or perceived violence of the participant.

Based on the distribution of data, neighborhood cohesion and violence were categorized into tertiles (Cohesion: T1 5 to 14, T2 15 to 16, T3 17 to 25; Violence: T1 5 to 7, T2 8 to 10, T3 11 to 20).

## Outcomes

The outcomes of these analyses were perceived stress, depressive symptoms, and hypertension, as established CVD risk factors [4, 5].

For the outcomes, stress was measured through the Perceived Stress Scale 4 (PSS-4) [21], a shortened 4- question version of the original 14 question Perceived Stress Scale [22]. Answer choices range from 0 = Never to 4 = Very Often. Although the PSS-4 has moderate loss in internal reliability versus the PSS-14 (r = 0.6 versus r = 0.85), its brevity is useful to settings with time constraints [21]. Our adaptation included all questions from the PSS-4, with a total score calculated by tabulating individual questions, for a total score ranging from 0 to 16. Cronbach's alpha was 0.37.

Depressive symptoms were measured through the Patient Health Questionnaire 9 (PHQ 9) [23], which consists of 9 questions related to symptoms of depression, and answer choices ranging from 0 = Not at all to 3 = Nearly every day. We retained all original questions, and calculated a total score by tabulating individual questions, for a total range from 0 to 27. Depression scores were further categorized as no depression ($\leq$5), mild (6 to 10), moderate (11 to 15), moderately severe (16 to 20), and severe depression ($\geq$21) [23]. The PHQ-9 is widely used in clinical and research contexts with high validity and reliability, and in our sample had a Cronbach's alpha of 0.78 [23].

Blood pressure (BP) measurements were taken using rigorous American Heart Association and World Health Organization guidelines [24, 25]. Semi-automated electronic cuffs (OMRON HEM 907) were used. Participants rested for five minutes, and then had three BPs measured, separated by 1-minute intervals. The average of the last 2 BPs was the BP used for analysis. BP categorizations were based on WHO guidelines [25]. Normal BP was defined as SBP < 120 mmHg and DBP < 90 mmHg. Prehypertension was defined as SBP 120–139 mmHg and DBP 80–89 mmHg. Hypertension was defined as SBP $\geq$ 140 mmHg, or DBP $\geq$ 90mmHg, or self-report of taking antihypertensive medication.

## Statistical analysis

For descriptive analyses, all scores were kept as continuous variables to allow comparison to other studies and summarized with median and interquartile range (IQR: 25th to 75th percentile) given non-normal distributions. Density plots were used to visualize the distributions.

For inferential analyses, linear regressions were used to examine the association between age categories and perceived neighborhood cohesion or violence as continuous scores. Unadjusted linear regression was used, with scatter plots, to examine the relationship between the exposures (perceived neighborhood cohesion, perceived neighborhood violence) and the continuous outcomes (stress, depressive symptoms). Finally, separate multivariable linear regression models or Poisson regression models with robust standard errors were used to examine the association between exposures and continuous or categorical outcomes, respectively, adjusting for age, sex, body mass index, smoking, alcohol, physical activity, diet, income and education. Stress was analyzed as a continuous outcome, while depression was analyzed as a categorical outcome (moderate to severe depression vs none or mild). Hypertension was also

analyzed as a categorical outcome (hypertension vs none). These analyses were conducted on the entire sample as presented in the main text, and sex-stratified as presented in the supplemental files.

As a sensitivity analysis, multivariable Poisson regression was used to examine the association between exposures and prehypertension as a categorical outcome, excluding people with hypertension.

Because temporal increases in neighborhood violence might have affected study enrollment, this could lead to selection bias (participants presenting for the study in more violent weeks are different from the general population). To explore this hypothesis, we conducted a sub-analysis comparing violence scores between low enrollment weeks (potentially due to higher violence) and high enrollment weeks. Weeks with enrollment less than the median of 22 enrollments per week were classified as low enrollment, while weeks with greater than 22 enrollments were classified as high enrollment. Violence scores between these two groups were compared with the non-parametric Wilcoxon rank sum test.

All analyses were conducted in R version 4.1.2.

## Ethics

This study was approved by institutional review boards at Weill Cornell Medicine and GHESKIO. Written informed consent was obtained from all participants.

## Results

A total of 2,961 participants were included in this analysis, with 58.0% female, and a median age of 40 years (Table 1). A third of participants had only primary level education or lower (35.7%), and the majority had a daily employment income of 1 US dollar or less (70.1%).

### Perceived neighborhood cohesion and violence

Participants reported high perceived neighborhood cohesion (median score 15 out of 25-point scale) with a relatively narrow IQR 14–17, and moderate perceived neighborhood violence (median score 9 out of 20-point scale) (Table 2, Fig 2A and 2B).

Older participants reported slightly higher perceived neighborhood cohesion (18–29 years 15.1, 60+ years 15.5; difference +0.3 score, 95% CI 0.09 to 0.5) and slightly lower perceived violence (18–29 years 9.4, 60+ years 8.7; difference -0.7 score, 95% CI -0.9 to -0.4).

### Stress, depression, and hypertension

The distribution of perceived stress was narrow, clustered around the median of 8, on a 16-point scale (IQR 6,10) (Fig 2C). Depressive symptoms were right skewed (Fig 2D), with 60.8% having no depression, 26.7% having mild depression, 8.8% moderate, 3.1% moderately severe, and 0.7% severe depression. For blood pressure, 23.4% of the cohort had a SBP $\geq$ 140 mmHg, 16.1% had a DBP $\geq$ 90 mmHg, and 29.2% had hypertension (Table 2, Fig 2E).

### Association between perceived neighborhood cohesion and mental or physical health outcomes

In unadjusted linear regression, perceived neighborhood cohesion was associated with lower stress at tertile 3 (T3 vs T1–0.45 score, 95% CI -0.72 to -0.18), and after multivariable adjustment, it remained associated with lower stress (T3 vs T1–0.41 score, 95% CI -0.68 to -0.15) (Table 3). In sex-stratified analyses, this relationship was seen only among males (T3 vs T1– 0.54 score, 95% CI -0.94, -0.14) (Table D in S1 Text).

**Table 1. Sociodemographic characteristics of Haitian adults in the Haiti CVD cohort study (N = 2961).**

| | Total population (N = 2961) |
|---|---|
| **Age, years** | **N (%)** |
| Median (IQR) | 40 [28, 55] |
| 18–29 | 872 (29.4) |
| 30–39 | 561 (18.9) |
| 40–49 | 526 (17.8) |
| 50–59 | 493 (16.6) |
| 60+ | 509 (17.2) |
| **Female** | 1718 (58.0) |
| **Education** | |
| Primary or lower | 1058 (35.7) |
| Secondary or higher | 1903 (64.3) |
| **Works for pay** | 982 (33.2) |
| **Income (daily)** | |
| $\leq$1 USD / day | 2075 (70.1) |
| >1 USD / day | 886 (29.9) |
| **BMI, kg/m$^2$** | |
| Underweight/Normal <24.9 | 1674 (56.5) |
| Overweight 25.0–29.9 | 773 (26.1) |
| Obese $\geq$30.0 | 512 (17.3) |
| **Smoking status** | |
| Never/Former | 2851 (96.4) |
| Current | 107 (3.6) |
| **Alcohol intake** | |
| $\leq$1 drink a day | 2846 (96.4) |
| >1 drink a day | 107 (3.6) |
| **Fruit/Vegetable intake** | |
| <5 servings a day | 2939 (99.3) |
| $\geq$5 servings a day | 20 (0.7) |
| **Physical activity** | |
| Low | 1498 (50.7) |
| Moderate-high | 1458 (49.3) |

IQR = Interquartile range.

In unadjusted Poisson regression, perceived cohesion was associated with lower prevalence ratio of moderate-to-severe depression with participants in Tertile 2 versus Tertile 1 having an increased PR of 0.96 (95% CI 0.94 to 0.99). After adjustment it remained associated with a lower prevalence ratio of depression (T2 vs T1 PR 0.97, 95% CI 0.94 to 0.99). In sex-stratified analyses, the lower prevalence of depression was only seen in females (Table D in S1 Text).

Perceived cohesion was not associated with hypertension in unadjusted, adjusted, or sex-stratified analyses. In a sensitivity analysis, perceived cohesion was associated with a lower prevalence ratio of prehypertension (T3 vs T1 PR 0.96, 95% CI 0.93 to 1.00) (Table B in S1 Text).

## Association between perceived neighborhood violence and mental or physical health outcomes

In unadjusted linear regression, perceived violence was not associated with stress (T3 vs T1 +0.23 score, 95% CI -0.32 to 0.50). After multivariable adjustment, perceived violence was

**Table 2. Neighborhood factors, stress, depression and blood pressure in the Haiti CVD cohort study.**

| Neighborhood cohesion | |
|---|---|
| Range | 5–25 |
| Median [IQR] | 15 [14, 17] |
| **Neighborhood violence** | |
| Range | 5–20 |
| Median [IQR] | 9 [7, 11] |
| **Perceived Stress Scale** | |
| Range | 0–16 |
| Median [IQR] | 8 [6, 10] |
| **Depression** | |
| Range | 0–27 |
| Median [IQR] | 4 [2, 8] |
| none (0–5) | 1799 (60.8) |
| mild (6–10) | 790 (26.7) |
| moderate (11–15) | 260 (8.8) |
| moderately severe (16–20) | 91 (3.1) |
| severe depression (21–27) | 21 (0.7) |
| **Hypertension** | |
| SBP $\geq$ 140 mmHg or DBP $\geq$ 90 mmHg or on medication | 864 (29.2) |

associated with higher stress (T3 vs T1 +0.29 score, 95% CI 0.01 to 0.56) (Table 4). In sex-stratified analyses, higher stress was only seen in females (T3 vs T1 + 0.38 score, 95% CI 0.02 to 0.74) (Table E in S1 Text).

Perceived violence was associated with a higher prevalence ratio of moderate-to-severe depression without adjustment (T3 vs T1 PR 1.14, 95% CI 1.11 to 1.18). After adjustment, perceived violence was associated with higher prevalence of depression (T3 vs T1 PR 1.12, 95% CI 1.09 to 1.16). This was true for both females and males in sex-stratified analyses (Table E in S1 Text).

Perceived violence was associated with lower hypertension in unadjusted analysis (T3 vs T1 prevalence ratio 0.74, 95% CI 0.64 to 0.87), but not after multivariable adjustment (T1 vs T1 prevalence ratio 0.99, 95% CI 0.87 to 1.13). The absence of this association was also seen in sex-stratified analyses (Table E in S1 Text). Perceived violence was not associated with prehypertension (Table C in S1 Text).

## Sub-analysis on perceived violence and enrollment

Weeks with low enrollment (< 22) had a median perceived violence score of 9 (IQR 7–11), and weeks with high enrollment ($\geq$ 22) had a median perceived violence score of 9 (IQR 7–11) (Fig B in S1 Text). There was no statistically significant detectable difference in violence between weeks with low versus high enrollment (Wilcoxon rank sum test p value = 0.20).

## Discussion

This data is among the first to describe perceived neighborhood cohesion and neighborhood violence in a population-based cohort of adults in Port-au-Prince, Haiti. We found high levels of perceived neighborhood cohesion (median 15/25) coupled with moderate levels of perceived violence (9/20). We found perceived neighborhood cohesion was associated with a small magnitude of lower stress and depressive symptoms. Perceived neighborhood violence

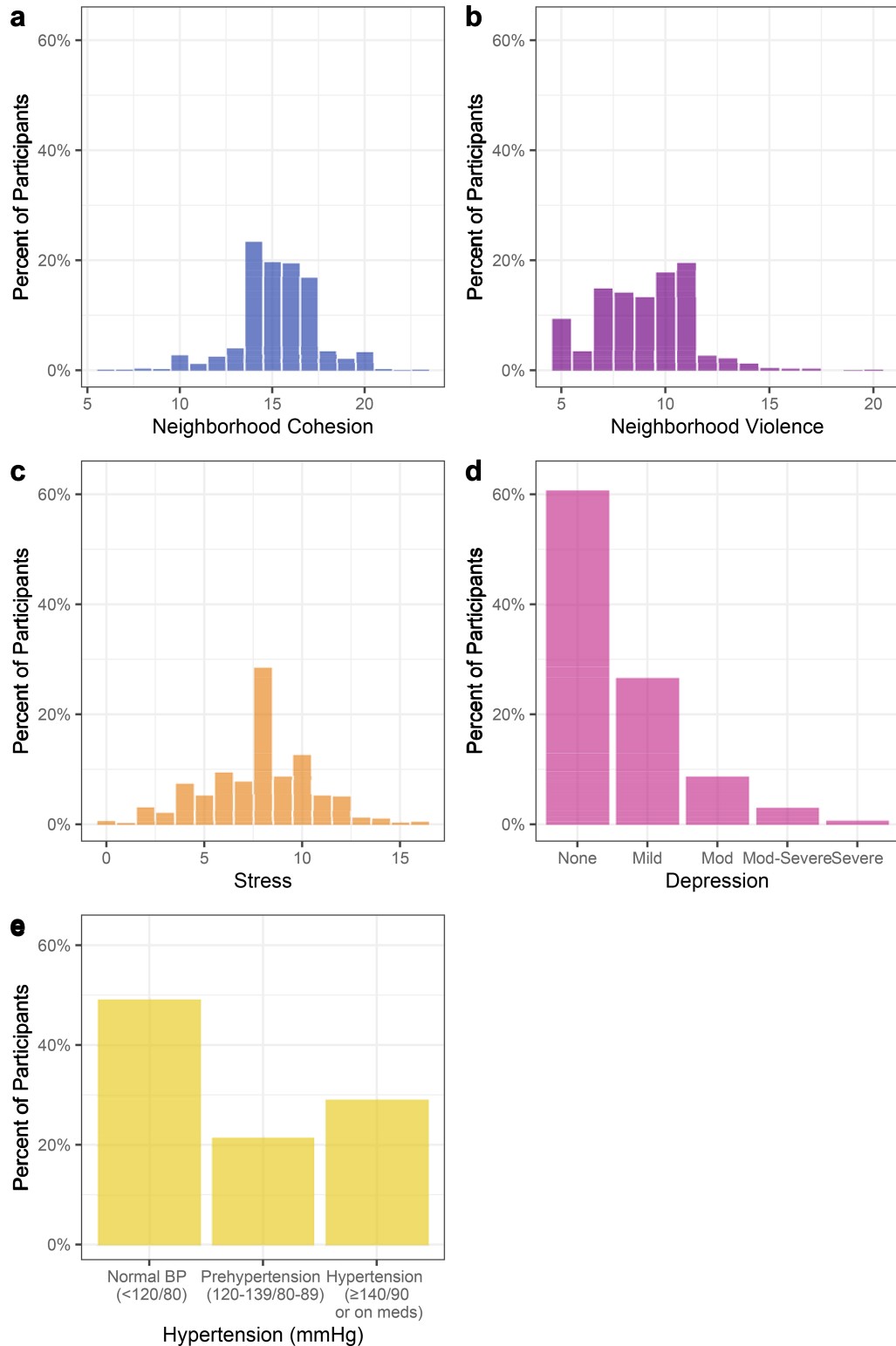

**Fig 2. Distribution of neighborhood cohesion, neighborhood violence, perceived stress, depression, and hypertension.** X axis represents the scores, or blood pressure categories. Y axis represents the percent of participants with that score.

**Table 3. Association between neighborhood cohesion and mental or physical health outcomes, multivariable linear regression and Poisson regression.**

| | Model 1: Stress | | Model 2: Moderate to Severe Depression | | Model 3: HTN | |
|---|---|---|---|---|---|---|
| | Beta [95% CI] | p | Prevalence Ratio [95% CI] | p | Prevalence Ratio [95% CI] | p |
| Neighborhood Cohesion Tertile 1 | ref | | ref | | ref | |
| Neighborhood Cohesion Tertile 2 | 0.15, [-0.07, 0.38] | 0.17 | 0.97 [0.94, 0.99]* | 0.01 | 1.01, [0.90, 1.13] | 0.87 |
| Neighborhood Cohesion Tertile 3 | -0.41, [-0.68, -0.15]* | <0.001 | 0.99 [0.96, 1.02] | 0.45 | 1.00, [0.88, 1.13] | 0.98 |
| **Age, years** | | | | | | |
| 18–29 | ref | | ref | | ref | |
| 30–39 | 0.13, [-0.18, 0.44] | 0.4 | 0.99 [0.96, 1.02] | 0.54 | 5.77, [3.52, 9.46] * | <0.001 |
| 40–49 | 0.41, [0.10, 0.72] * | 0.01 | 1.01 [0.98, 1.05] | 0.45 | 13.2, [8.26, 21.1] * | <0.001 |
| 50–59 | 0.08, [-0.26, 0.42] | 0.64 | 0.99 [0.95, 1.02] | 0.43 | 19.6, [12.3, 31.4] * | <0.001 |
| 60+ | 0.47, [0.13, 0.81] * | 0.01 | 1.02 [0.98, 1.06] | 0.44 | 24.7, [15.5, 39.4] * | <0.001 |
| **Male vs Female** | -0.80, [-1.01, -0.58] * | <0.001 | 0.89 [0.86, 0.91]* | <0.001 | 1.03, [0.92, 1.14] | 0.63 |
| **Income (daily)** | | | | | | |
| ≤1 USD / day | 0.25, [0.04, 0.46] * | 0.02 | 1.08 [1.06, 1.11]* | <0.001 | 1.06, [0.95, 1.18] | 0.31 |
| >1 USD / day | ref | | ref | | ref | |
| **Education** | | | | | | |
| Primary or lower | ref | | ref | | ref | |
| Secondary or higher | -0.54, [-0.78, -0.29] * | <0.001 | 0.98 [0.95, 1.01] | 0.14 | 0.81, [0.72, 0.91] * | <0.001 |
| **BMI, kg/m$^2$** | | | | | | |
| Underweight, Normal, Overweight (<29.9) | ref | | ref | | ref | |
| Obese (≥30.0) | -0.24, [-0.50, 0.03] | 0.08 | 0.98 [0.95, 1.01] | 0.24 | 1.26, [1.12, 1.42] * | <0.001 |
| **Smoking** | | | | | | |
| Never/Former | ref | | ref | | ref | |
| Current | 0.38, [-0.10, 0.86] | 0.12 | 1.04 [0.98, 1.11] | 0.18 | 0.82, [0.61, 1.11] | 0.2 |
| **Alcohol intake** | | | | | | |
| ≤1 drink a day | ref | | ref | | ref | |
| >1 drink a day | 0.34, [-0.25, 0.94] | 0.26 | 1.100 [1.03, 1.17]* | <0.001 | 0.95, [0.64, 1.39] | 0.78 |
| **Physical activity** | | | | | | |
| Low | -0.32, [-0.52, -0.12] * | <0.001 | 0.93 [0.91, 0.95]* | <0.001 | 1.10, [0.99, 1.21] | 0.07 |
| Moderate-high | ref | | ref | | ref | |
| **Fruit Vegetable Daily Intake** | | | | | | |
| <5 servings | ref | | ref | | ref | |
| ≥5 servings | 0.57, [-0.32, 1.45] | 0.21 | 1.04 [0.89, 1.20] | 0.63 | 0.74, [0.31, 1.77] | 0.5 |

* significant at p < 0.05. Stress range 0–16. Depression range 0–27. Multivariable linear regression was used for the outcome of stress. Thus, in these models the beta coefficient represents the unit change in the outcome for each tertile of neighborhood cohesion, compared to tertile 1. Poisson regression was used for the outcomes of moderate-to-severe depression and hypertension. The exponentiated beta coefficient is the prevalence ratio of the outcome for each tertile of neighborhood cohesion, compared to tertile 1.

was associated with a small magnitude of higher stress, and a large magnitude of higher depressive symptoms. Neither cohesion nor violence were associated with hypertension, although higher cohesion was associated with lower ratios of prehypertension.

We unexpectedly found that perceived neighborhood violence was lower than we hypothesized, given the rising social and political turbulence in Port-au-Prince during the study period [26]. Historically, Haiti has a history of political instability and violence rooted in slavery, French colonialism, and foreign interference [14]. This legacy manifests today as weak institutional systems and ongoing political instability during the study period that include large political protests, gang activity, kidnappings, and violence that especially affect urban areas

**Table 4. Association between neighborhood violence and mental or physical health outcomes, multivariable linear regression and Poisson regression.**

| | Model 1: Stress | | Model 2: Moderate to Severe Depression | | Model 3: HTN | |
|---|---|---|---|---|---|---|
| | Beta [95% CI] | p | Prevalence Ratio [95% CI] | p | Prevalence Ratio [95% CI] | p |
| Neighborhood Violence Tertile 1 | ref | | ref | | ref | |
| Neighborhood Violence Tertile 2 | 0.00, [-0.22, 0.22] | 0.99 | 1.05 [1.02, 1.07]* | <0.001 | 0.93, [0.83, 1.04] | 0.23 |
| Neighborhood Violence Tertile 3 | 0.29, [0.01, 0.56] * | 0.04 | 1.12 [1.09, 1.16]* | <0.001 | 0.99, [0.87, 1.13] | 0.85 |
| **Age, years** | | | | | | |
| 18–29 | ref | | ref | | ref | |
| 30–39 | 0.12, [-0.19, 0.42] | 0.45 | 0.99 [0.95, 1.02] | 0.41 | 5.78, [3.53, 9.47] * | <0.001 |
| 40–49 | 0.40, [0.09, 0.72] * | 0.01 | 1.01 [0.98, 1.05] | 0.41 | 13.2, [8.28, 21.1] * | <0.001 |
| 50–59 | 0.08, [-0.26, 0.42] | 0.65 | 0.99 [0.96, 1.03] | 0.69 | 19.6, [12.3, 31.3] * | <0.001 |
| 60+ | 0.47, [0.13, 0.81] * | 0.01 | 1.02 [0.98, 1.06] | 0.37 | 24.7, [15.5, 39.5] * | <0.001 |
| **Male vs Female** | -0.85, [-1.06, -0.63] * | <0.001 | 0.89 [0.87, 0.91] | <0.001 | 1.30, [0.92, 1.14] | 0.62 |
| **Income (daily)** | | | | | | |
| ≤1 USD / day | 0.17, [-0.04, 0.39] | 0.12 | 1.07 [1.04, 1.09] | <0.001 | 1.07, [0.95, 1.19] | 0.27 |
| >1 USD / day | ref | | ref | | ref | |
| **Education** | | | | | | |
| Primary or lower | ref | | ref | | ref | |
| Secondary or higher | -0.54, [-0.79, -0.30] * | <0.001 | 0.97 [0.94, 1] | 0.07 | 0.81, [0.72, 0.92] * | <0.001 |
| **BMI, kg/m$^2$** | | | | | | |
| Underweight, Normal, Overweight (<29.9) | ref | | ref | | ref | |
| Obese (≥30.0) | -0.25, [-0.51, 0.02] | 0.07 | 0.98 [0.95, 1.01] | 0.19 | 1.26, [1.12, 1.41] * | <0.001 |
| **Smoking** | | | | | | |
| Never/Former | ref | | ref | | ref | |
| Current | 0.43, [-0.06, 0.92] | 0.09 | 1.04 [0.98, 1.1] | 0.21 | 0.83, [0.61, 1.11] | 0.21 |
| **Alcohol intake** | | | | | | |
| ≤1 drink a day | ref | | ref | | ref | |
| >1 drink a day | 0.26, [-0.33, 0.85] | 0.38 | 1.08 [1.01, 1.15] | 0.02 | 0.94, [0.64, 1.38] | 0.76 |
| **Physical activity** | | | | | | |
| Low | -0.26, [-0.46, -0.07] * | 0.01 | 0.94 [0.92, 0.96]* | <0.001 | 1.09, [0.99, 1.21] | 0.08 |
| Moderate-high | ref | | ref | | ref | |
| **Fruit Vegetable Daily Intake** | | | | | | |
| <5 servings | ref | | ref | | ref | |
| ≥5 servings | 0.64, [-0.18, 1.46] | 0.12 | 1.06 [0.92, 1.23] | 0.42 | 0.74, [0.31, 1.77] | 0.5 |

* significant at p < 0.05. Stress range 0–16. Depression range 0–27. Multivariable linear regression was used for the outcome of stress. In this model the beta coefficient represents the unit change in the outcome for each tertile of neighborhood violence, compared to tertile 1. Poisson regression was used for the outcomes of moderate-to-severe depression and hypertension. The exponentiated beta coefficient is the prevalence ratio of the outcome for each tertile of neighborhood violence, compared to tertile 1.

including Port-au-Prince [26]. There are a few possible reasons for the discrepancy between the perceived neighborhood violence in this study and the observed violence during the study period. One possibility is that participants interpreted the boundaries of their "neighborhood" very narrowly. The Haitian Creole questionnaire on violence asked specifically if participants had relatives or friends that had been the victim of violence. While almost all participants knew of someone who was the victim of theft or an attack, they would only answer yes to the question if their direct family or friends were affected. Secondly, perhaps participants felt that if they reported violence, they would somehow be responsible or implicated in the activities. Lastly, Haitian culture is rooted in resiliency and hope, which may influence the way

participants answered questions around violence. Many participants report faith as a source of hope, and report hesitancy in verbalizing violence, which would be seen as capitulating to negative circumstances. At the same time, perceived cohesion was high among study participants. While the Neighborhood Collective Efficacy instrument has not been used previously in Haiti to our knowledge, high levels of support have been reported in other studies among Haitian adolescents and survivors of the 2010 earthquake [16, 27]. Qualitative research is needed to elucidate reasons behind these findings.

Participants also reported moderate levels of perceived stress and depression. We found 13% of patients had a PHQ-9 $\geq$10, higher than the 8.1% of US adults from 2013–2016 in the National Health and Nutrition Examination Survey [28]. The other available studies from Haiti center around mental health symptoms after the 2010 earthquake in Haiti, with one meta-analysis finding 32.2% reported severe symptoms of depression [29]. While severe depression appears to have decreased over time, there are persistent levels of stress and depression beyond the earthquake. Secondly, even though reported violence is lower than expected, perceived stress and depression are not.

We found perceived neighborhood violence was associated with a higher prevalence ratio of moderate-to-severe depression and a small magnitude of perceived stress, while we found perceived neighborhood cohesion was associated with a small magnitude of lower stress and lower prevalence of depression. Prior work suggests social cohesion is protective against depression, and exposure to violence is associated with higher depression [2]. Neighborhood social cohesion may result in informal support systems, leading to improved health behaviors. Higher social cohesion has been linked with higher antihypertensive medication adherence [30] and even lower CVD incidence and mortality in Sweden [31, 32], but the external validity of this finding outside of a high-income country with a robust welfare system is not certain. In the US Multi-Ethnic Study of Atherosclerosis, social cohesion was not associated with incident hypertension [33]. In sex-stratified analyses, the strongest association between perceived violence and higher depression was seen in both females and males. The remaining associations were seen only in one biological sex. In males, perceived cohesion was associated with lower stress. In females, perceived cohesion was associated with lower depression and perceived violence with higher stress. This may suggest sex-specific mechanisms for how environmental factors affect mental health outcomes.

In terms of physical health outcomes, there was no association between perceived violence, perceived cohesion and hypertension, although perceived cohesion was associated with lower prehypertension. One explanation is that perceived cohesion or violence contribute to subclinical disease, but the association is not seen with clinical disease of hypertension after accounting for basic demographic factors. Another explanation is there may have been omitted variable bias, or residual confounding with age, given the statistically significant relationship between older age and lower perceived violence scores. Either participants in violent areas are not surviving to older ages due to the violence, or older people in violent areas are more likely to move away than younger people. We did not find evidence of selection bias on perceived violence and enrollment.

Strengths of this analysis include its design as a population-based cohort in an urban LMIC setting, use of validated instruments for perceived neighborhood cohesion and violence, and research-grade measurement of blood pressure. Limitations include the potential selection bias due to civil unrest, use of self-reported data, the cross-sectional study design, and potential omitted variable bias in the regressions such as lack of data on sleep patterns.

In conclusion, we found high perceived neighborhood cohesion and moderate perceived neighborhood violence in one of the most unstable countries in the world. We found perceived neighborhood cohesion was associated with lower stress and depression, while perceived

violence was associated with a higher prevalence ratio of moderate-to-severe depression, and a small magnitude of higher stress. We did not find an association between these neighborhood effects and hypertension. Our findings underscore the fact that perceived neighborhood context may differ from observed political and civil events in complex environments such as Haiti and that ongoing qualitative research is needed to better understand perceived neighborhood effects that may contribute to CVD related and other health outcomes.

## Supporting information

**S1 Text.**
(DOCX)

## Acknowledgments

We thank the study participants, and the study staff, in particular the psychologists and community health workers.

## Author Contributions

**Conceptualization:** Lily D. Yan, Margaret L. McNairy, Monica M. Safford, Vanessa Rouzier.

**Data curation:** Lily D. Yan, Jean Lookens Pierre, Eliezer Dade, Rodney Sufra, Nicholas Roberts, Stephano St Preux, Miranda Metz.

**Formal analysis:** Lily D. Yan, Linda M. Gerber, Olga Tymejczyk.

**Funding acquisition:** Margaret L. McNairy, Denis Nash, Monica M. Safford, Jean W. Pape.

**Investigation:** Jean Lookens Pierre, Eliezer Dade, Rodney Sufra, Jean W. Pape.

**Methodology:** Lily D. Yan, Jessy G. Dévieux, Linda M. Gerber, Stephano St Preux, Rodolphe Malebranche, Olga Tymejczyk, Denis Nash.

**Project administration:** Jean Lookens Pierre, Eliezer Dade, Miranda Metz, Marie Deschamps, Jean W. Pape, Vanessa Rouzier.

**Resources:** Stephano St Preux, Marie Deschamps, Jean W. Pape.

**Software:** Lily D. Yan.

**Supervision:** Margaret L. McNairy, Jessy G. Dévieux, Vanessa Rouzier.

**Writing – original draft:** Lily D. Yan.

**Writing – review & editing:** Lily D. Yan, Margaret L. McNairy, Jessy G. Dévieux, Jean Lookens Pierre, Eliezer Dade, Rodney Sufra, Linda M. Gerber, Nicholas Roberts, Stephano St Preux, Rodolphe Malebranche, Miranda Metz, Olga Tymejczyk, Denis Nash, Marie Deschamps, Monica M. Safford, Jean W. Pape, Vanessa Rouzier.

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
