## [Decision Letter · Decision Letter 0]

8 Apr 2022

PGPH-D-22-00428

Neighborhood cohesion and violence in Port-au-Prince, Haiti, and their relationship to stress, depression, and hypertension: findings from the Haiti Cardiovascular Disease Cohort Study

Dear Dr. Yan,

Thank you for submitting your manuscript to PLOS Global Public Health. After careful consideration, we feel that it has merit but does not fully meet PLOS Global Public Health’s publication criteria as it currently stands. Therefore, we invite you to submit a revised version of the manuscript that addresses the points raised during the review process.

We look forward to receiving your revised manuscript.

Kind regards,

Maurizio Trevisan, M.D., MS

Academic Editor

Journal Requirements:

2. Your co-authors:

Margaret L McNairy -mam9365@med.cornell.edu

Jessy G Dévieux -devieuxj@fiu.edu

Jean Lookens Pierre -jlookens@gheskio.org

Eliezer Dade -eliezerdade@gheskio.org

Rodney Sufra -rsufra@gheskio.org

Stephano St Preux -stpreuxstephano@gheskio.org

Rodolphe Malebranche -r_malebranche@yahoo.fr

Denis Nash -r_malebranche@yahoo.fr

Marie Deschamps -mariehd@gheskio.org

Monica M Safford -mms9024@med.cornell.edu

Jean W Pape -jwpape@gheskio.org

Vanessa Rouzier -vrouzier@gheskio.org

,have not confirmed authorship of the manuscript. We have resent them the authorship confirmation email; however please check that the above email address for them is correct and follow up personally to ensure they confirm. 

3. Please amend your detailed Financial Disclosure statement. This is published with the article, therefore should be completed in full sentences and contain the exact wording you wish to be published.

a). State the initials, alongside each funding source, of each author to receive each grant.

4. Please provide us with a direct link to the base layer of the map used in Fig 1 and ensure this location is also included in the figure legend. 

Please note that, because all PLOS articles are published under a CC BY license (creativecommons.org/licenses/by/4.0/), we cannot publish proprietary maps such as Google Maps, Mapquest or other copyrighted maps. If your map was obtained from a copyrighted source please amend the figure so that the base map used is from an openly available source.

Please note that only the following CC BY licences are compatible with PLOS licence: CC BY 4.0, CC BY 2.0  and CC BY 3.0, meanwhile such licences as CC BY-ND 3.0 and others are not compatible due to additional restrictions. If you are unsure whether you can use a map or not, please do reach out and we will be able to help you. 

The following websites are good examples of where you can source open access or public domain maps:

5. We have noticed that you have uploaded supporting information but you have not included a list of legends.  Please add a full list of legends for all supporting information files (including figures, table and data files) after the references list.

Additional Editor Comments (if provided):

Reviewers' comments:

Reviewer's Responses to Questions

**Comments to the Author**

1. Does this manuscript meet PLOS Global Public Health’s publication criteria? Is the manuscript technically sound, and do the data support the conclusions? The manuscript must describe methodologically and ethically rigorous research with conclusions that are appropriately drawn based on the data presented.

Reviewer #1: Yes

Reviewer #2: Yes

2. Has the statistical analysis been performed appropriately and rigorously?

Reviewer #1: No

Reviewer #2: Yes

3. Have the authors made all data underlying the findings in their manuscript fully available (please refer to the Data Availability Statement at the start of the manuscript PDF file)?

Reviewer #1: Yes

Reviewer #2: Yes

4. Is the manuscript presented in an intelligible fashion and written in standard English?

Reviewer #1: Yes

Reviewer #2: Yes

5. Review Comments to the Author

Reviewer #1: Comments

1. I am not sure depression variable is appropriate for the linear regression analysis. Figure 2d is showing skewed distribution. I am suggesting authors to consider the logistic regression analyses using known cutoff points for all the outcome variables (stress, depression, hypertension).

2. Page 5 line 90, Authors described that the cohort study participants are selected by a multistage random sampling. Does it mean that cohort members represent entire residents of Port-au-Prince? More explanation is regarding sampling methods is needed instead of just citing another paper (ref 18). In addition, is there any kind of sampling weights for each study participants?

3. Neighborhood cohesion and violence seems to be negatively correlated and can be evaluated concurrently instead of independently. How about making a new variable based on the cohesion and violence information and evaluate whether there is any interaction between two variables. For example, study participants could be divided into participants with low cohesion + low violence (ref); participants with low cohesion + high violence; participants with high cohesion + low violence; participants with high cohesion + high violence.

4. Detailed explanations regarding adjusted variables are needed. For example, how physical activities are categorized into low and moderate and high? What is low and high physical activities? How are servings defined? What is the difference between primary and secondary education is Haiti?

5. Page 7 line 135: Detailed cutoff values should be presented.

6. Table 3 4 Considering very small amount of discussion regarding effects of adjusted variables on outcome variables, I am not sure coefficient for age, education, sex, BMI, and other confounder variables should be presented in the main Table.

Reviewer #2: Overall, this is an interesting study and well written manuscript, based on cross-sectional data from a population-based sample of adults in Port-au-Prince, Haiti. The authors examined cross-sectional associations of neighborhood cohesion and violence with a range of outcomes including perceived stress, depression, and hypertension prevalence.

Given the relatively unexplored setting, the present study may provide some novel evidence from an unstable, transitional setting, such as Haiti. They found higher violence associated with higher stress and depression, in line with previous evidence. There was no consistent association with hypertension. The relatively large sample size and random sampling are major strengths of the study. There are a few concerns, which are outline below. Specifically:

1) Please clarify the participation and response rate of the random sample to rule out major selection bias in the study population.

2) The percentage of missing data seems very low (only 1.5%), which is surprising. Can you please clarify the strategies implemented to mitigate this potential issue?

3) Hypertension was defined based on blood pressure values and self-report of medication. What about self-report of physician diagnosis?

4) I would strongly encourage the authors to examine the same associations with prehypertension as an alternative outcome, which might provide additional novelty to the study.

5) Likewise, I would strongly encourage the authors to provide sex-stratified analyses, in line with current trends in cardiovascular epidemiology.

6) The authors often talk in terms of high or moderate neighborhood cohesion and violence. It is unclear the reference/range used to define these two parameters as high, moderate or low. Is this classification in relation to previous studies? Please clarify this issue.

7) Among the study limitations, the authors should also acknowledge the cross-sectional design, as well as the lack of additional potential confounders, such as sleep patterns, which have been linked to both neighborhood characteristics and CVD risk.

6. PLOS authors have the option to publish the peer review history of their article (what does this mean?). If published, this will include your full peer review and any attached files.

**Do you want your identity to be public for this peer review?** For information about this choice, including consent withdrawal, please see our Privacy Policy.

Reviewer #1: No

Reviewer #2: **Yes: **Saverio Stranges

---

## [Decision Letter · Decision Letter 1]

16 Jun 2022

Neighborhood cohesion and violence in Port-au-Prince, Haiti, and their relationship to stress, depression, and hypertension: findings from the Haiti Cardiovascular Disease Cohort Study

PGPH-D-22-00428R1

Dear Dr Yan,

We are pleased to inform you that your manuscript 'Neighborhood cohesion and violence in Port-au-Prince, Haiti, and their relationship to stress, depression, and hypertension: findings from the Haiti Cardiovascular Disease Cohort Study' has been provisionally accepted for publication in PLOS Global Public Health.

Best regards,

Maurizio Trevisan, M.D., MS

Academic Editor

Reviewer Comments (if any, and for reference):

Reviewer's Responses to Questions

**Comments to the Author**

1. If the authors have adequately addressed your comments raised in a previous round of review and you feel that this manuscript is now acceptable for publication, you may indicate that here to bypass the “Comments to the Author” section, enter your conflict of interest statement in the “Confidential to Editor” section, and submit your "Accept" recommendation.

Reviewer #2: All comments have been addressed

2. Does this manuscript meet PLOS Global Public Health’s publication criteria? Is the manuscript technically sound, and do the data support the conclusions? The manuscript must describe methodologically and ethically rigorous research with conclusions that are appropriately drawn based on the data presented.

Reviewer #2: Yes

3. Has the statistical analysis been performed appropriately and rigorously?

Reviewer #2: Yes

4. Have the authors made all data underlying the findings in their manuscript fully available (please refer to the Data Availability Statement at the start of the manuscript PDF file)?

Reviewer #2: Yes

5. Is the manuscript presented in an intelligible fashion and written in standard English?

Reviewer #2: Yes

6. Review Comments to the Author

Reviewer #2: The authors have adequately addressed the comments raised in a previous round of review

7. PLOS authors have the option to publish the peer review history of their article (what does this mean?). If published, this will include your full peer review and any attached files.

**Do you want your identity to be public for this peer review?** For information about this choice, including consent withdrawal, please see our Privacy Policy.

Reviewer #2: **Yes: **SAVERIO STRANGES
